# An Alternative Technology to Obtain Dewatered Mine Tailings: Safe and Control Environmental Management of Filtered and Thickened Copper Mine Tailings in Chile

**Carlos Cacciuttolo [1],\* and Edison Atencio [2,3]**

[1]   Civil Works and Geology Department, Catholic University of Temuco, Temuco 4780000, Chile
[2]   School of Civil Engineering, Pontificia Universidad Católica de Valparaíso, Av. Brasil 2147, Valparaíso 2340000, Chile
[3]   Department of Management, Economics, and Industrial Engineering, Politecnico di Milano, Via Lambruschini 4b, Bovisa, 20156 Milan, Italy
\*   Correspondence: ccacciuttolo@uct.cl or carlos.cacciuttolo@gmail.com

**Abstract:** An alternative process to obtain a high degree of dewatering tailings that produces a high-density product is presented in this article. This technology involves the combination of tailings particle grain size classification by hydrocyclones (HC) and tailings dewatering by horizontal vibratory screens (HVS). It makes it possible to dewater tailings to a high grade of performance. This alternative technology (HC-HVS) involves the recovery of water from the coarse fraction of tailings (sands) through two hydrocycloning stages, followed by a dewatering stage of cycloned tailings sands on horizontal vibratory screens, to reduce moisture content and turn it into a "cake". The resulting coarse fraction tailings are easily transported to a dry stack tailings storage facility (TSF). The fine fraction of tailings (slimes) can be dewatered on thickener equipment to recover part of the process water. Finally, this article describes the main benefits of this alternative dewatered tailings technology with an emphasis on (i) dewatering technology evolution over the last 17 years; (ii) process stages features; (iii) pilot test results; (iv) tailings properties analysis (such as particle grain size distribution, fines content) and (v) lessons learned about the experience gained in the operation of Mantos Blancos case study with dry stack tailings storage facility.

**Keywords:** filtered tailings; vibratory dewatering screens; hydrocyclones; dry stack tailings storage facility; tailings dewatering; tailings cake

## 1. Introduction

### 1.1. Efficient Use of Water and Safe Management in Mine Tailings

Engineers, scientists, mine operators, and authorities are working to improve the design and operations of tailings storage facilities (TSF) in areas of safe storage and efficient water management [1]. Concerning this last issue, i.e., water management, the effort has been focused on the development of optimal solutions, which consider the following: (i) reliable performance of technologies; (ii) a dynamic water balance (keeping in mind the specific conditions of the site); and (iii) an efficient management of water with emphasis on the control of water losses (evaporation and seepage) [1–3]. If these key aspects are achieved successfully, such as (i) a decrease in the requirement for freshwater (make-up), (ii) a decrease in the negative environmental impacts (seepages, dust emission, groundwater pollution, among others), and (iii) an increase in the opportunity for others to use the natural water for other uses (urban zones, agricultural zones, environment, among others) sustainable development will be reached. Therefore, the resulting approach to water management will promote an advance in the development of sustainable technologies for the mining industry [1,2,4–6].

The application of tailings dewatering technologies to increase water reclaim is an important step in decreasing water losses (water coming from freshwater supplies or the sea in some cases) [2,4,7–9]. Water losses occur because of evaporation, infiltration, and water retention in the pores of the tailings in Tailings Storage Facilities (TSF) [1,2]. This challenge requires new designs for tailings management, and such designs must make tailings management less negative impacts on the environment and use water more efficiently [4,10]. Considering the Best Available Technologies (BATs) for tailings management incorporating tailings dewatering techniques is possible to mention four main categories: (i) Conventional Tailings, (ii) Thickened Tailings, (iii) Paste Tailings, and (iv) Filtered Tailings [2,4,11,12].

Conventional copper tailings typically range from 25%–40% solids weight concentrations (Cw), thickened copper tailings 40%–65% (Cw), paste copper tailings 65%–80% (Cw), and filtered copper tailings over 80% (Cw) (solid concentrations may vary with particle size and shape, clay content, mineralogy, electrostatic forces, and flocculant dosing). Conventional, thickened, paste and filtered tailings refer to a continuum of tailings with high solid concentrations and higher yield stress due to greater fluid removal from tailings before disposal [2,11].

In current Chilean large-scale mining in dry climate areas, most typical tailings disposal schemes consist of conventional or slightly thickened at modest levels of tailings solids weight concentration (Cw 25%–40%). Conventional TSFs have dams built of the coarse fraction of tailings (cycloned tailings sand) obtained by hydrocyclones or have slightly thickened tailings deposits with dams built of borrowed material. Conventional tailing dams may have water recoveries as high as 65%–75% in well-operated TSFs, which means they have appropriate tailings distribution, good control of the pond (volume and location), and adequate seepage recovery. In conventional dams, water at the settling pond is decanted by floating pumps or decant towers, and dam seepages are collected by drainage and cutoff trench systems. However, a high seasonal evaporation rate can substantially reduce water recovery from the pond area, and infiltration from the pond in contact with natural soil can produce water losses. Some mining operations with this technology in Chile are Chuquicamata, El Soldado, Los Pelambres, and Los Bronces [2,11].

Thickened Tailings Disposal (TTD) technology requires more background data than conventional tailings disposal. In the conventional approach, the properties of tailings are fixed by the concentrator plant. In contrast, in a TTD tailings storage facility, the properties of the tailings and their placement are "engineered" to suit the topography of the disposal area. The behavior of tailings in the two approaches is entirely different. In conventional disposal, tailings segregate as they flow and settle out to an essentially flat deposit, whereas in TTD technology, a sloping surface is obtained. The principal difference is that, in TTD technology, tailings are thickened before discharge to a homogeneous heavy consistency, resulting in a laminar non-segregating flow. In this way, TTD produces high water recovery (70% of tailings water recovery) and a self-supporting deposit with sloping sides, requiring small dams. Some mining operations with this technology in Chile are Centinela (Esperanza), Spence, and Sierra Gorda [2,11].

Paste Tailings Technology has been applied on a small production scale because equipment manufacturing ability is limited. This method permits obtaining a medium make-up water requirement (80% of tailings water recovery). However, in some cases, there are difficulties in tailings transportation requiring the use of positive displacement pumping (PD Pumps), resulting in the highest capital/operating costs. The main advantages of this method are that: large dams are not required; Only small dams are needed, and the emission of dust is negligible because the surface of the deposited tailings remains as a solid hard crust due to the bonding of the tailings particles by the action of the flocculant. Some mining operations with this technology in Chile are Planta Demo Collahuasi, Delta Plant Enami, Coemin, Las Cenizas, and Alhue [2,11,13].

In the last 20 years, many mining projects worldwide have applied a tailings disposal technology called dry stacking of filtered tailings. This technique produces an unsaturated cake that allows storing this material without the need to manage large slurry tailings

ponds. The application of this technology has accomplished: (i) an increase in water recovery from tailings (90% of tailings water recovery), (ii) a reduction of TSF footprint (impacted areas), and (iii) a decrease in the risk of physical instability because TSFs are self-supporting structures under compaction (such as dry stacks), and (iv) a better community perception. Some mining operations with this technology are: Salares Norte, Potrerillos, San Jose Plant Pucobre, El Gato, Tambo de Oro, Huasco, La Coipa, El Peñon, and Mantos Blancos [2,12,14–20].

The risk associated with TSFs is very large, considering that a potential dam failure could easily damage the environment and people with pollution or other adverse impacts [21,22]. Accidents or failures at these facilities have always been associated with social, environmental, or public safety issues, some more catastrophic than others [21,23]. The above has contributed to a reduction in the rate of TSF failures in recent decades. In most countries around the world where responsible mining is practiced daily, regulatory frameworks have become more stringent, requiring a higher level of responsibility, with the intent of minimizing risks posed to society and the environment [4,10,24].

Filtered tailings are an environmentally friendly technology since it generates negligible seepages, significantly decreasing the risk of contaminants' transportation [12,25–29]. However, a relevant feature to consider is to avoid dust or fine particulate entrainments carried by local winds throughout the life of the TSF. Dust emission controls will play an important role as good design and proper implementation will provide the primary control mechanism for dust following regulatory air quality requirements [12,25–29]. Some dust control alternatives are soil cover (borrowed material), topsoil/revegetation cover, phytostabilization, binder material, or chemical agglomeration [12].

Filtered TSF offers progressive reclamation and mine closure activities during the TSF operation, controlling dust with cover material placement, TSF side slope reclamation, and revegetation (if required) such as using the phytostabilization technique. The combination of these technologies decreases TSF seepage to very low levels, almost avoiding tailings leachate and reducing the risk of geochemical contamination. Under dry climate conditions, no supernatant TSF ponds need to be monitored during operation, and no freeboard requirement of tailings dams, reducing water losses significantly and eliminating an overtopping or piping (internal dam erosion by seepage) event risk, respectively [12].

A filtered tailings TSF solution is very competitive today compared to conventional technologies because the cost of closure and post-closure are lower, the environmental and construction permits are easier to obtain, and the authorities and community are more prone to accept it [12].

From 1998 to date there have been major disasters worldwide due to tailings storage facility failures. Some examples are Los Frailes Aznalcóllar Spain (1998), Baia Mare Romania (2000), Kolontar Hungary (2010), Mount Polley Canada (2014), Fundao Samarco Brazil (2015), Corrego de Feijao Brumandinho Brazil (2019) and Jagersfontain South Africa (2022), which have caused the death of hundreds of people and irreparable environmental damage [21–24]. As a consequence of these disasters, the international community, institutions, and global groups such as the ICMM (International Council on Mining and Metals), the UN environment program, and the PRI (Principles for Responsible Investments), developed a "Global Tailings Management Standard for the Mining Industry" launched in August 2020, to regulate the operation throughout the entire life cycle of tailings storage facilities, including closure and post-closure (perpetuity), considering zero harm to people and the environment, and zero tolerance for human fatalities [30]. ICMM advocates that the application of appropriate design and management standards and good practices allow tailings storage facilities to be safe. ICMM members are committed to preventing catastrophic failures of tailings deposits with continuous improvement in these storage facilities' design, construction, and operation stages. This organization urges mining companies to improve their management by adopting the Global Industry Standard on Tailings Management, taking advantage of technological innovation and continuous improvement [30].

In this context, the current regulatory framework needs reassessment to consider the best available practices and technologies for tailings storage and eliminate tailings dam failures worldwide to establish safe tailings management by implementing environmentally friendly solutions [1,6,31].

### 1.2. Aim of the Article

This article presents a new alternative process to obtain a high degree of dewatering tailings. It is an emerging combination of dewatering tailings technologies that produce a high-density product. This technology combines tailings particle grain size classification by Hydrocuclones (HC) and tailings dewatering by Horizontal Vibrating Screens (HVS). The HC-HVS technology (combination of particle grain size classification tailings with the use of HC and dewatering of the tailings with the use of HVS) has been used in the last 17 years (2005–2022) in the dewatering of copper mine tailings. Dewatering vibratory screens were initially used to filter industrial fine-size minerals, such as silica, feldspar, kaolin, fluorite, salts, carbon, etc. This dewatering was done to allow for storage in silos and decrease the storage volumes [32,33]. In this manner, the optimal management of the products known as bulk solids was obtained, both in hoppers and conveyor belts.

In Chile, dewatering vibratory screens have been used mainly to treat carbon, silica sands for fusion and glass, rinse salts in the chemical industry, and the aggregates and fine minerals obtained from mining processing [33,34].

The experience developed in the North of Chile has enabled the dewatering of tailings to a high degree and the production of a copper tailings product with a low moisture content, which facilitates the disposal of tailings in the environment as a tailings storage facility in a dry stockpile.

This article describes the main benefits of this alternative dewatering tailings technology (HC-HVS) with an emphasis on (i) dewatering technology evolution over the last 17 years; (ii) process stages features; (iii) pilot test results; (iv) tailings properties analysis (particle grain size distribution, fines content, density, permeability, etc.); and (v) lessons learned about the experience gained in the operation of a case study in Chile called Mantos Blancos in a dry stack tailings storage facility.

### 1.3. Description of Mantos Blancos Dewatered Tailings Management-Study Case

The Mantos Blancos copper sulfide plant, located in the Atacama Desert, Chile, started operation in 1981 [35]. It has had a tailings treatment plant from its inception. It consists of a tailings classification circuit that thickens the fine fraction of the tailings and filters the coarse fraction. Since this plant is located in one of the world's driest areas, the coarse tailings were filtered to maximize the water recovery. Disc filters were used in the beginning. Later in 1984, during the expansion of the concentrator plants to a production of 8000 mtpd, vacuum belt filters were incorporated to dewater the coarse tailings, which were in use until 2005 (Figure 1). The tailings were finally deposited in a TSF with a dam to impound thickened slimes and in a dry stack with a dike to retain the coarse fraction in a 40/60% cut, respectively, and a solids concentration of 60% (Cw) and 80% (Cw), respectively [32].

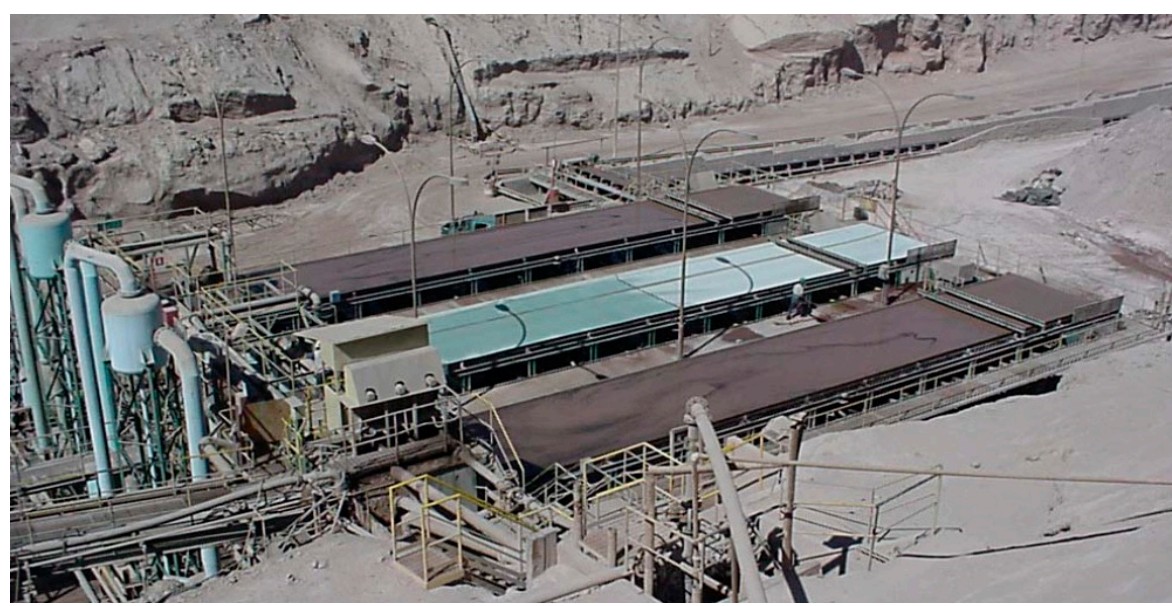

**Figure 1.** Tailings vacuum Belt Filters in Mantos Blancos Project [32].

The copper tailings from the concentrator plant were classified in a hydrocyclone station of 500 mm Eral Hydrocyclones, functioning by gravity, in a conical configuration, with four operating units and two on standby. This classification system produced a 50%–55% solids distribution and a cut size of $d_{50}$ between 50–60 microns. The overflow of the hydrocyclones, solid concentration 24%–26% (Cw), was transported to three thickening units (Larox of 60 m diameter, Dorr Oliver of 45 m diameter, and Eimco of 45 m diameter) for the process of sedimentation, using a dosage of flocculent of 5 g/ton, that later feeds with a solids content of 55%–60% (Cw) the filtration units and the fine tailings dam in a ratio restricted by the capacity of absorption of fines in the filters. The thickeners recovered approximately 63 to 67% of water, which was returned to the concentrator plant (See Figure 2) [32].

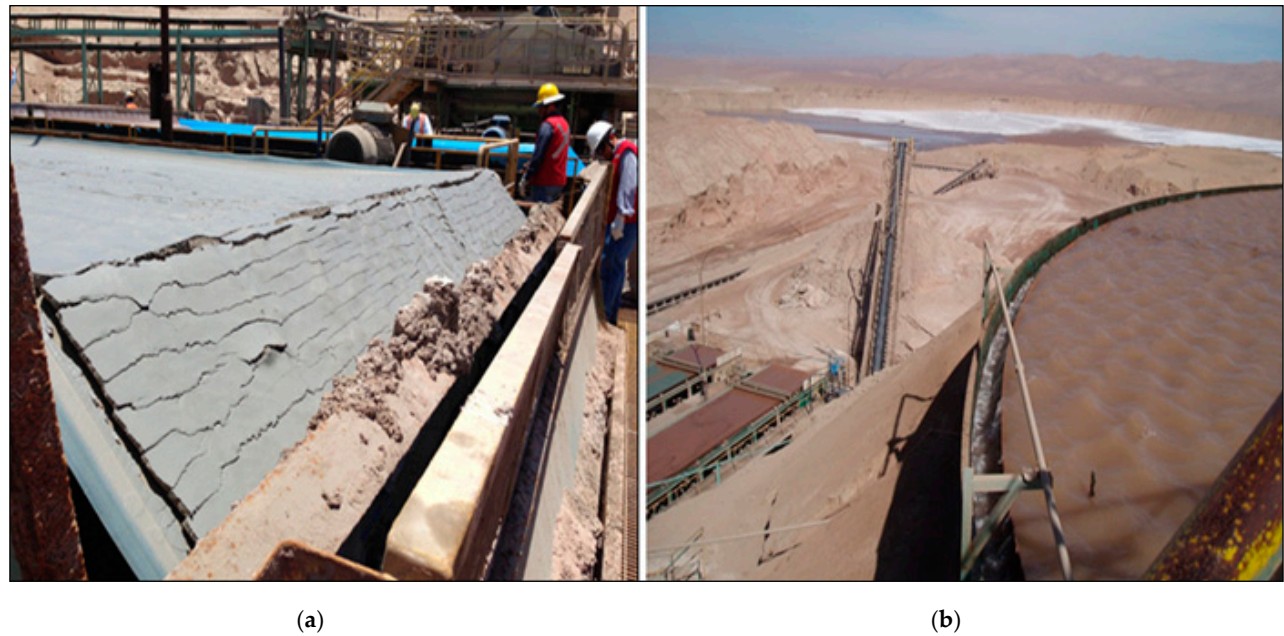

(**a**)          (**b**)

**Figure 2.** Dewatering of Mantos Blancos Coarse Tailings with Vacuum Belt Filters and Slimes Thickener Overview (**a**) Filtered Coarse Tailings with Vacuum Belt Filters, (**b**) Slimes Thickening Overview.

The underflow of the hydrocyclones, with a solids content of 65%–67% (Cw), was sent to three filtration units (band filters of 100 m² each) and was previously mixed with parts of thickened fine tailings. Coarse tailings cakes were obtained with a 17% moisture content considering wet basis (equivalent to a 23% moisture content considering dry basis), which were later transported to the dry stack TSF of coarse material by conveyor belts. The filtered water, plus the wash water of the filtering cloth, was recirculated to the thickeners to sediment solids in suspension.

From the beginning, the objective of the classification and dewatering of the tailings was to decrease the amount of water in the tailings management. Later, in 2004 and 2005, the concept of replacing the vacuum belt filters with vibratory screens was analyzed, starting a pilot plant, driven mainly by the need to reduce operational and maintenance costs.

## 2. Materials and Methods

### 2.1. Process Description of Tailings Dewatering by HC-HVS Technology

An emerging technology (HC-HVS) is presented that makes it possible to dewater tailings to a high grade of performance, which involves the recovery of the coarse fraction of tailings (sands) through two hydrocycloning stages followed by a drainage stage or dewatering of cycloned tailings sands on horizontal vibratory screens, to reduce moisture content and turn it into filtered tailings or "cake" [36]. The resulting material is easy to transport to the adjoining dry stack TSF. The fine fraction of tailings (slimes) can be dewatered on high-density thickener equipment to recover part of the process water.

This technology's operational principle is forming a fine layer of tailings on the screen. This way, the filtering or dewatering process is carried out mainly via a stratified tailings material bed, which is in contact with the HVS. Although the opening of the screen in its lower parts may be larger than the smallest particles of the tailings being processed, the formation of a filtering bed on the upper part of the stratified tailings material achieves an efficient decrease of the moisture of the dewatered tailings. Also, it allows for a significant reduction in the loss of fine tailings [36]. The formation of this bed is done with hydrocyclones that recover the filter material on the HVS (water and fine tailings). These are the real pillars of the tailings dewatering with HVS; this alternative process consists of the recovery of solids > 45 microns content in the tailings (See Figure 3) [34,36].

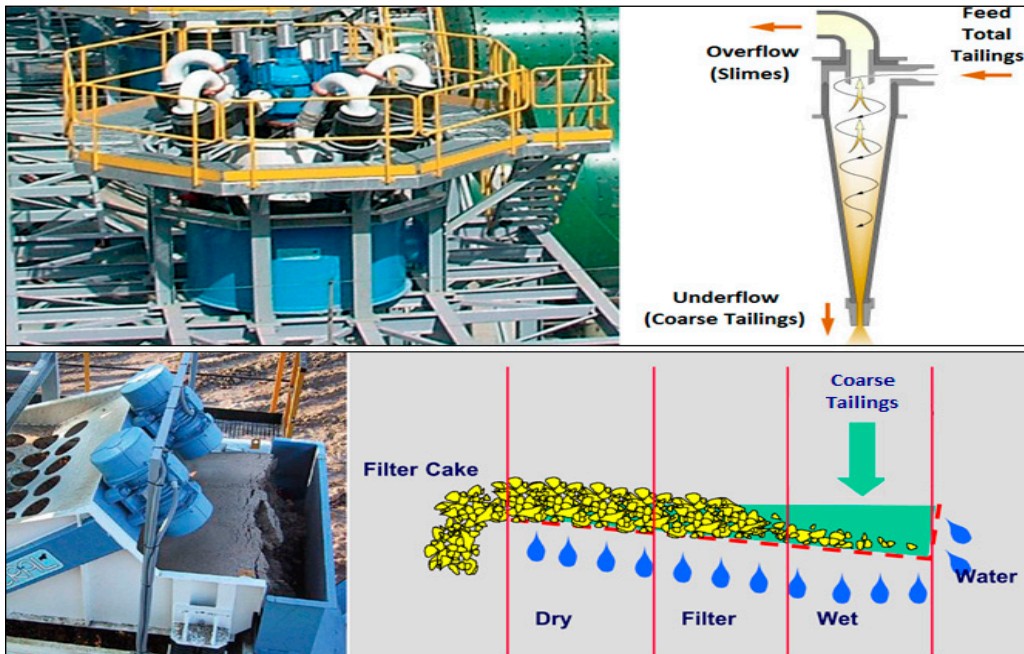

**Figure 3.** Schematic View of Dewatering Process on Hydrocyclone (HC) + Horizontal Vibratory Screen (HVS) (Adapted figure from [37]).

Figure 4 presents a typical process diagram of Mantos Blancos dewatering copper tailings with the HC-HVS technology. This diagram shows that the total tailings generated by the concentrator plant are classified in Hydrocyclone Station No. 1, to produce a fine fraction of the tailings (slimes) and a fraction of coarse tailings (sands). The coarse fraction of the tailings (underflow of the cyclones) is sent to the HVS [38].

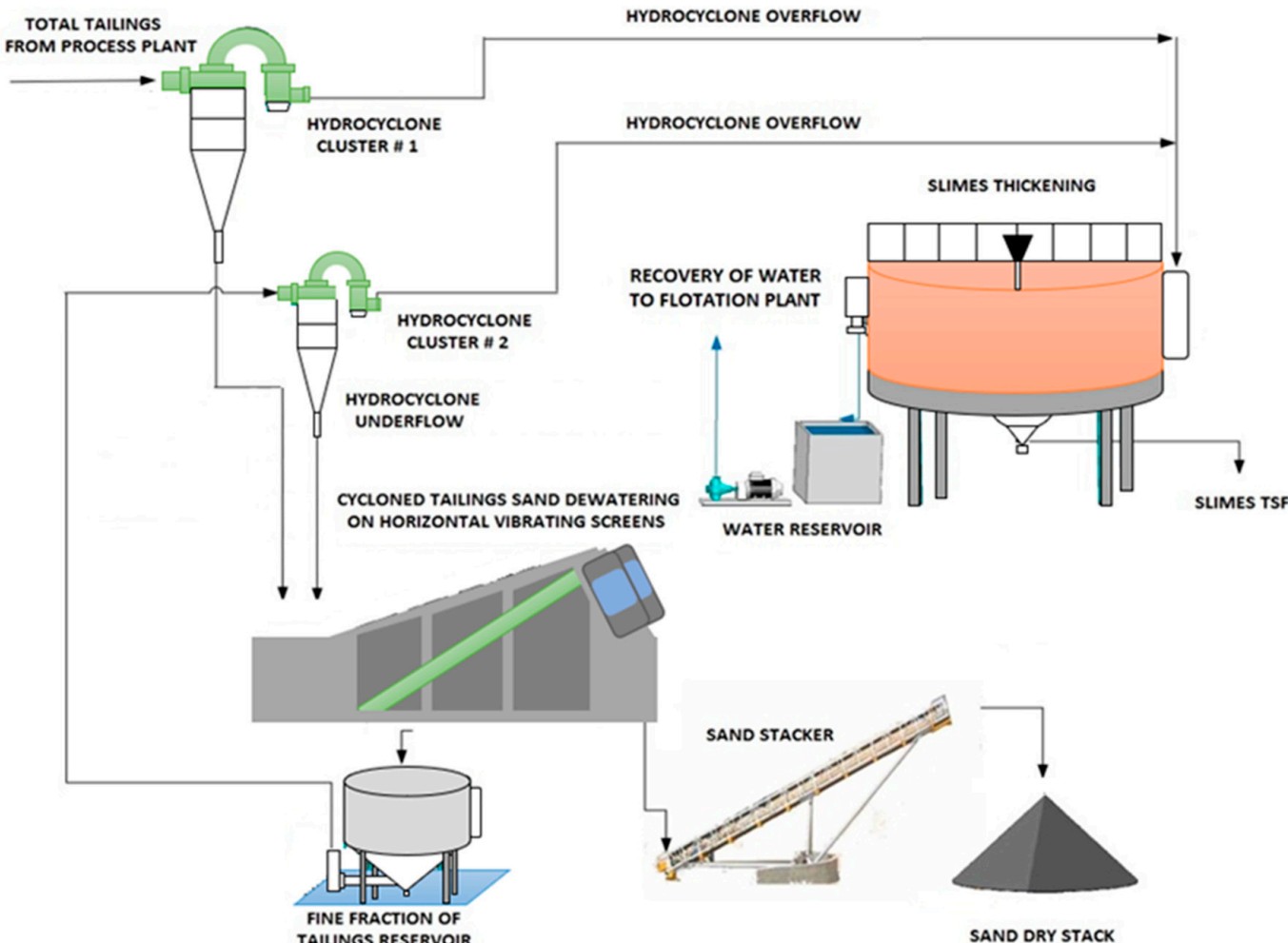

**Figure 4.** Copper tailings dewatering process with HC-HVS technology-Process Tailings Flowsheet (Adapted figure from [38]).

The coarse fraction of the tailings dewatered in the HVS is discharged to a conveyor belt to be transported to the TSF and deposited by a Mobile Conveyor Stacker and Tripper. The fine fraction of tailings obtained in the HVS, which has high water content, is sent to a box that feeds Hydrocyclone Station No. 2 located above the vibratory screen. Later, the coarse fraction of filtering obtained by Hydrocyclone Station No. 2 is discharged on the tailings "cake", which is located in the HVS. This "cake" is formed by the feed of the coarse fraction of the tailings resulting from the first classification stage using hydrocyclones. In this manner, the final tailings are filtered through this "cake", retaining the fines in the screen and preventing their filtration through the screen panels. The above is why this type of tailings dewatering plant can filter the solids to a size significantly finer than the opening of the screen installed in the HVS [32,38].

The overflow from Hydrocyclone Station No. 1 is sent to a fines thickener to recover water. This water is returned to the concentrator plant for reuse in the metallurgical process [32,38].

## 3. Results

### *3.1. Pilot Test Dewatering Application*

From the beginning, the objective of the classification and dewatering of the tailings was to decrease the amount of water in the tailings management. Later, in 2004–2005, the concept of replacing the vacuum belt filters with a vibratory screen was analyzed, driven mainly by the need to reduce operational and maintenance costs. A dewatering pilot plant was assembled, installing a compact stage of HC and an HVS to dewater the coarse fraction of the tailings [39].

### 3.1.1. First Stage Process Description-Tailings Particle Grain Size Classification

The tailings from the flotation plant were classified into two fractions within an existing hydrocyclone station (Hydrocyclone Station No. 1). The fine fraction was sent to the tailings thickeners as was previously done. A fraction of coarse tailings was sent to the new dewatering pilot plant and a new hydrocyclone station (Hydrocyclone Station No. 2), instead of being sent to the vacuum belt filters [39].

In order to recover a maximum of coarse particles, the existing 500 mm hydrocyclones (Hydrocyclone Station No. 1) were maintained in operation. However, their configuration and geometry were modified to increase the solids content at the discharge of the underflow. To do this, it was necessary to decrease the separation size, $d_{50}$ [34,39]. To achieve this objective, the future classification tailings with the 500 mm hydrocyclones were simulated with a 250 mm hydrocyclone, to obtain an appropriate amount of solids and thus accommodate the treatment capacity of the tailings dewatering pilot plant, installed as a second stage in the process [39].

### 3.1.2. Second Stage Process Description-Dewatering of Coarse Tailings

This dewatering stage consists of a compact hydrocyclone pilot plant, which includes a vibratory screen that directly receives the coarse fraction of the tailings obtained during the first stage of classification, and which produces a final "cake" of course tailings with sufficient concentration to be transported to the dry stack TSF with conveyor belts. The vibratory screen used in the pilot tests was located between the extremes of the vacuum belt filters and the conveyor belts [39].

The tailings slurry filtered by the vibratory screen is pumped to a hydrocyclone battery that performs a solid/liquid separation. Given that the objective of this classification stage (Hydrocyclone Station No. 2) is to recover almost all of the total solids in the dewatering operation, 100 mm hydrocyclones with a high shear capacity are used. The hydrocyclones produced overflow with a minimum solids content in weight, and this overflow is sent to the thickeners together with the fines obtained during the first stage of tailings classification.

The pilot plant used at Mantos Blancos (2004–2005), included the following equipment: (i) a compact hydrocyclone plant with two 100 mm hydrocyclones; (ii) a horizontal vibratory screen of 300 mm in width and 1600 mm in length; and (iii) a slurry centrifugal pump [39].

This equipment presents significant advantages in terms of a decrease of: (i) area; (ii) power consumption; (iii) wear elements (filter cloths); and (iv) reagents (flocculants). The result is a simple operating system for tailings dewatering [34,39]. Figure 5 presents the main equipment of the dewatering pilot plant used in the Mantos Blancos Project.

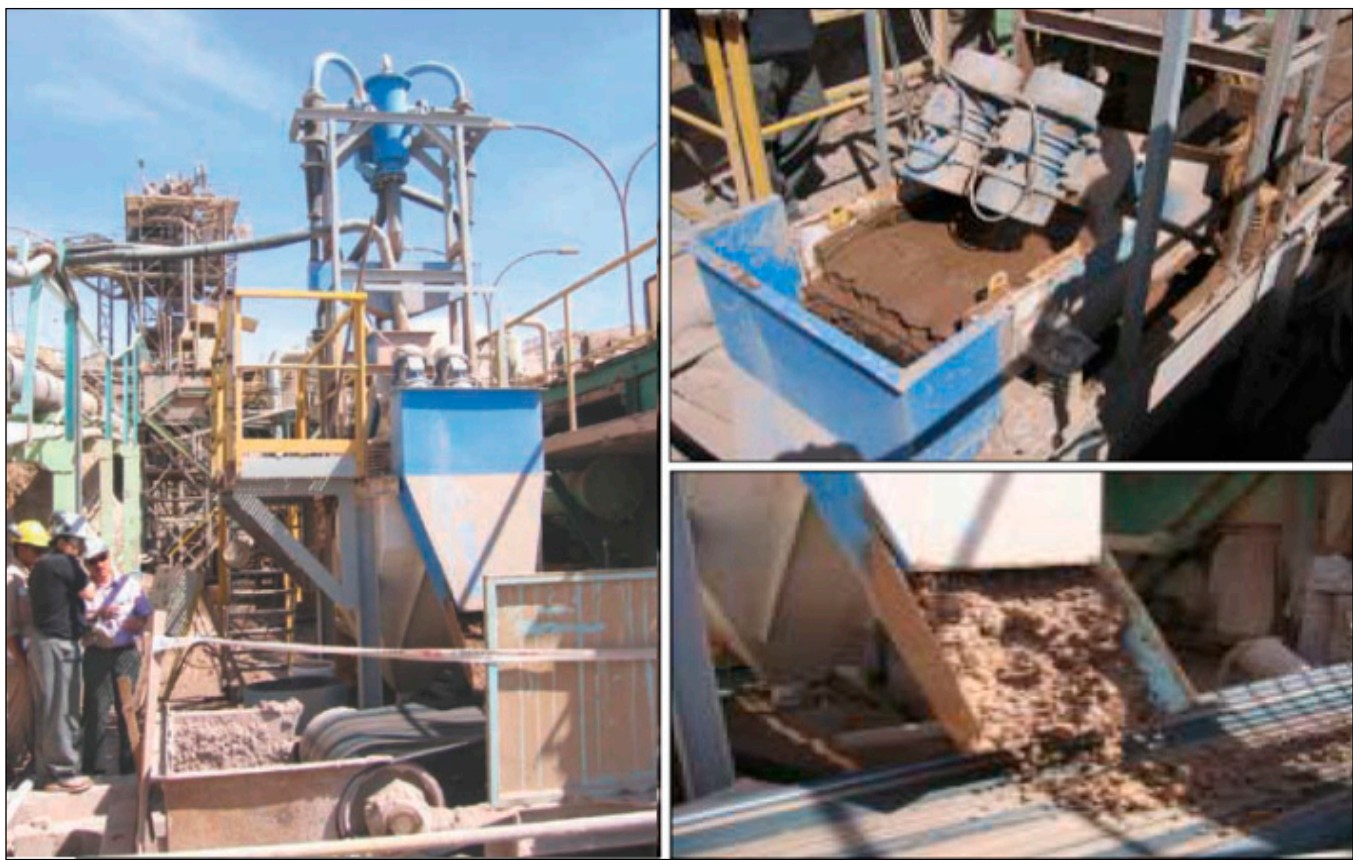

**Figure 5.** Dewatering Pilot Plant 600 mtpd: Hydrocyclone + Horizontal Vibratory Dewatering Screens Overview–Horizontal Vibratory Screens Operation–Cake Tailings Product [34].

### 3.1.3. Pilot Test Dewatering Application Results

Preliminary tests were done to provide the circuit's necessary stability to achieve the pilot plant's maximum efficiency. Adjustments were performed to the plant to obtain a "cake", not overly dry but with the maximum moisture allowed in tailings for the transport in conveyor belts [34,39].

According to the results obtained in the pilot tests with vibratory screens, in addition to a fine selection of the configuration of the hydrocyclones, it was possible in this case to replace part of the existing belt filters with a compact plant with HVS together with HC, with a lower operating cost. The tailings solids content was 56%, with 82% (Cw) for test No. 1 and 68% with 78% (Cw) for test No. 2; this indicates a high degree of dewatering of the tailings [39].

The pilot tests performed during this period showed that a two-stage classification system with hydrocyclones and one stage with a vibratory screen obtained the same solids content in weight as the belt filters. Accordingly, it was decided to install two units with a capacity to treat 50% of the coarse tailings. Considering these results, Mantos Blancos decided to perform industrial-scale trials [34,39].

### 3.2. First Industrial Trial Phase Application

The concept of using vibratory screens for tailings dewatering was undertaken to decrease the solids content in weight from 80% to 70% (Cw) of the coarse fraction of the tailings; the objective was to facilitate transport and disposal in the dry stack TSF. The coarse tailings with a solids content of 80% (Cw) transported to the TSF by a conveyor belt are stacked with steep slopes of 15%. By decreasing the solids content of the coarse tailings, coarse tailings flow further from the discharge point and occupy more surface area in the tailings storage facility [34,39].

### 3.2.1. First Stage Process Description-Tailings Particle Grain Size Classification

For the first stage of the hydrocyclones, the gravitational battery of 500 mm hydrocyclones (Hydrocyclone Station No. 1) was maintained. Only the configuration was changed to decrease the tailings split and to increase the solids fraction in the discharge, which is fed directly to the vibratory screen. The fine fraction of the tailings obtained from the overflow of the hydrocyclones was sent to the thickeners [34,39]. The first industrial unit used in Mantos Blancos (2007) included the following components described below, keeping in mind the first stage of the hydrocyclone:

Gravitational hydrocyclones battery, Model ERAL DEP-6-G4, with 02 hydrocyclones, Model PP050102 V, 500 mm in diameter, made of polyurethane.

### 3.2.2. Second Stage Process Description-Dewatering of Coarse Tailings

An HVS was installed to dewater the coarse fraction of the tailings. The fines that pass through the vibratory screen plus the filtered water are collected in a rectangular tank located underneath the vibratory screen, from which the suspension is pumped to the second stage of hydrocycloning (Hydrocyclone Station No. 2). Later, the fine tailings and filtered tailings are collected and recirculated to be placed on a pre-layer over the coarse tailings that are on the vibratory screen—this is done in order to dewater the resulting cake [34,39]. The first industrial unit used in Mantos Blancos (2007) included the following components described below:

Hydrocyclone battery, Model ERAL DEF-38-G4 with 38 hydrocyclones installed and 19 hydrocyclones in operation, Model PP010041 II, 100 mm in diameter, made of polyurethane.

A horizontal vibratory screen, Model EV-86, with a 38 kW installed power, with a functional area of filtering of 11.5 m$^2$.

Centrifugal pump group GB-86, with an 8/6 pump and 75 kW of installed power.

### 3.2.3. First Industrial Trial Phase Application Results

According to the results obtained from the first industrial phase, where part of the coarse tailings was dewatered with vibratory screens, in addition to performing an adequate selection of the configuration of the hydrocyclones, where 50% of the coarse tailings were dewatered with the use of vacuum belt filters and the rest is dewatered with the use of an HC-HVS, it is possible to conclude that in this case, it is feasible to replace all of the existing belt filters by a compact plant with vibratory screens and hydrocyclones, with a lower operating cost. It is relevant to mention that the solids content (Cw) at the discharge of the vibratory sieve was decreased from 80% to 70%, thus achieving the objective of facilitating their disposal in a dry stack TSF. With these results, Mantos Blancos decided to perform the second phase of industrial trials [34,38,39]. The following Table 1 and Figure 6 present the results of the first stage of the industrial trial and an overview of the dewatering plant:

**Table 1.** Process Performances and Results–1st Tailings Dewatering Industrial Plant (HC-HVS).

| First Stage–Tailings Classification (Hydrocyclone Cluster #1) | | | Second Stage–Tailings Dewatering (Vibratory Dewatering Screen + Hydrocyclone Cluster #2) | | |
|---|---|---|---|---|---|
| Tailings Flow at cluster #1 feed | 11,592 | mtpd | Tailings Flow at HVS feed | 4056 | mtpd |
| Solid content at cluster #1 feed | 41 | % | Solid content at cluster #2 feed | 35 | % |
| Solid content at cluster #1 overflow | 21 | % | Solid content at cluster #2 overflow | 5 | % |
| Solid content at cluster #1 underflow | 70 | % | Sand/slime split | 92/08 | - |
| Sand/slime split | 70/30 | - | Solid content at HVS discharge | 70 | % |
| Tailings Flow to underflow | 8112 | mtpd | Tailings Flow at HVS discharge | 3744 | mtpd |

Note: Only 50% of coarse tailings are dewatered in the HC-HVS compact plant at this trial phase.

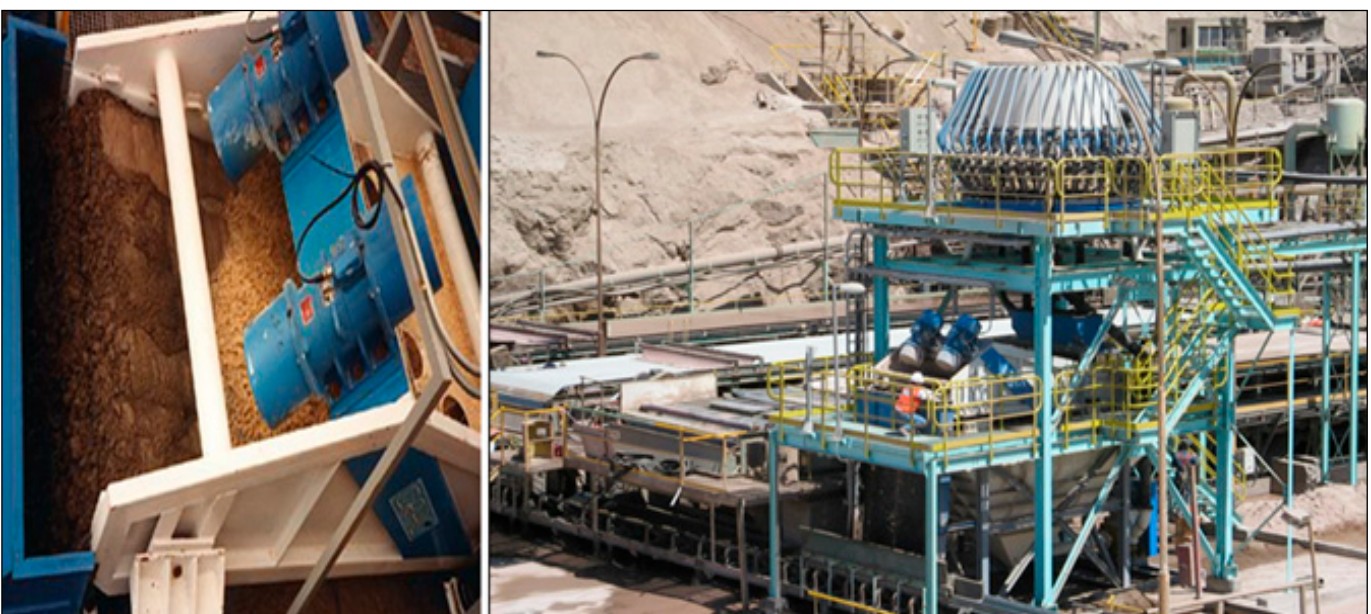

**Figure 6.** 1st Industrial Trial Phase of Mantos Blancos Tailings Plant 4056 mtpd: Hydrocyclone Cluster + 01 Horizontal Vibratory Dewatering Screen [40,41].

### 3.3. Second Industrial Trial Phase Application and Results

As a consequence of the good outcomes obtained with the first trials at an industrial scale, a second vibratory screen was installed in January 2010, with a capacity to dewater approximately 8112 mtpd of coarse tailings. The above replaced the tailings filtering activities performed by the vacuum belt filters. Conceptually, the tailings' primary grain size classification process in this second stage of trials at an industrial scale is the same as applied in the first stage of industrial trials, where only the number of hydrocyclones and HVSs used is changed [34,39].

### 3.3.1. First Stage Process Description-Tailings Particle Grain Size Classification

The quantity of hydrocyclones used is presented below:

Gravitational hydrocyclone cluster, Model ERAL DEP-6-G4, with 04 hydrocyclones, Model PP050102 V, 500 mm in diameter, made of polyurethane (PU).

### 3.3.2. Second Stage Process Description-Dewatering of Coarse Tailings

The number of hydrocyclones, HVSs, and pump equipment are presented below:

Hydrocyclone cluster, Model ERAL DEF-38-G4 with 38 hydrocyclones installed and all in operation, Model PP010041 II, of 100 mm in diameter, made of PU.

02 horizontal vibratory screens, Model EV-86, with a 38 kW installed power, with an area of filtering of 11.5 m$^2$.

Centrifugal pump group GB-86, with a 10/8 pump and 110 kW of installed power.

### 3.3.3. Second Industrial Trial Phase Application Results

The following Table 2 and Figure 7 present the results of the second stage of the industrial trials and an overview of the HC-HVS technology, respectively:

**Table 2.** Process Performances and Results–2nd Tailings Dewatering Industrial Plant (HC-HVS).

| First Stage–Tailings Classification (Hydrocyclone Cluster #1) | | | Second Stage–Tailings Dewatering (Vibratory Dewatering Screen + Hydrocyclone Cluster #2) | | |
| --- | --- | --- | --- | --- | --- |
| Tailings Flow at cluster #1 feed | 11,592 | mtpd | Tailings Flow at HVS feed | 8112 | mtpd |
| Solid content at cluster #1 feed | 41 | % | Solid content at cluster #2 feed | 35 | % |
| Solid content at cluster #1 overflow | 21 | % | Solid content at cluster #2 overflow | 5 | % |
| Solid content at cluster #1 underflow | 70 | % | Sand/slime split | 92/08 | - |
| Sand/slime split | 70/30 | - | Solid content at HVS discharge | 70 | % |
| Tailings Flow to underflow | 8112 | mtpd | Tailings Flow at HVS discharge | 7488 | mtpd |

Note: The operational availability of this HC-HVS compact plant at the 2nd industrial trial has been of the order of ~90%.

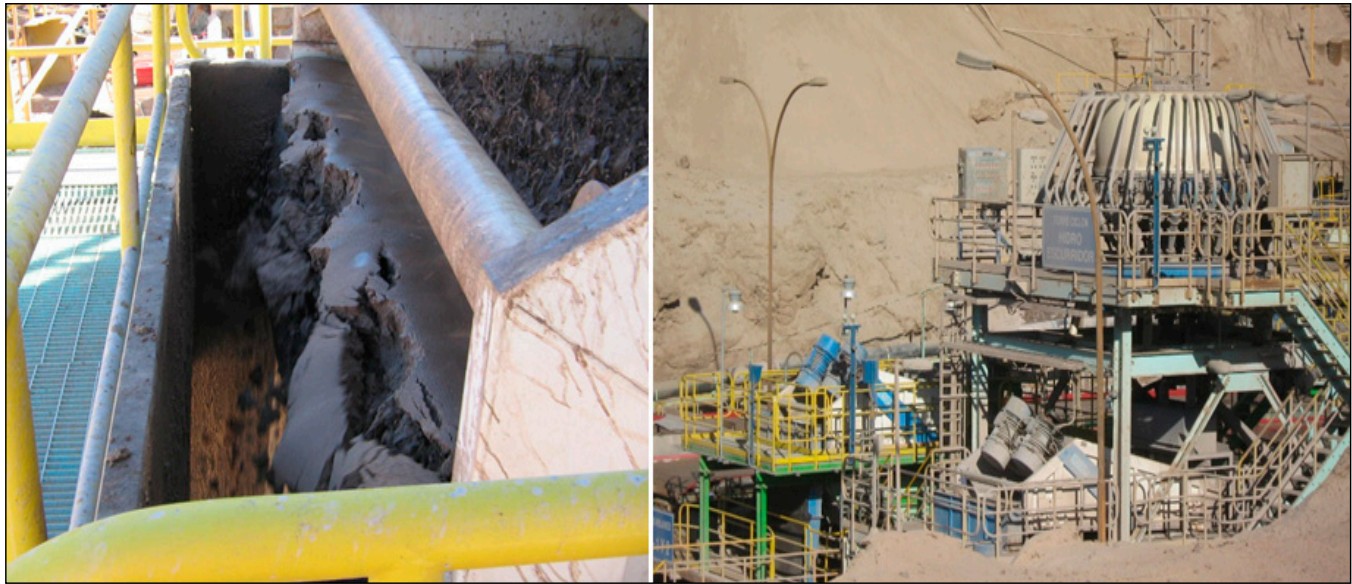

**Figure 7.** 2nd Industrial Trial Phase of Mantos Blancos Tailings Plant 8000 mtpd: Hydrocyclone Cluster + 02 Horizontal Vibratory Dewatering Screens [40,41].

Finally, the tailings produced with the use of the HC-HVS technology present the following characteristics:

- $P_{80}$: 255 to 270 μm
- Fines content: 23 to 25%
- Solids content (Cw): 70%

The operational availability of the coarse tailings dewatering plant for its equipment is equivalent to 90%.

### 3.4. Tailings Storage Facilities Results

The thickened fine tailings (slimes) are transported by pumping pipelines to a tailings storage facility near the tailings dewatering plant. In this place, the tailings are deposited through a series of spigots, forming a tailings beach and a clear water lagoon. The process waters are recirculated to the metallurgical process plant through pumping and pipelines, as shown in Figure 8.

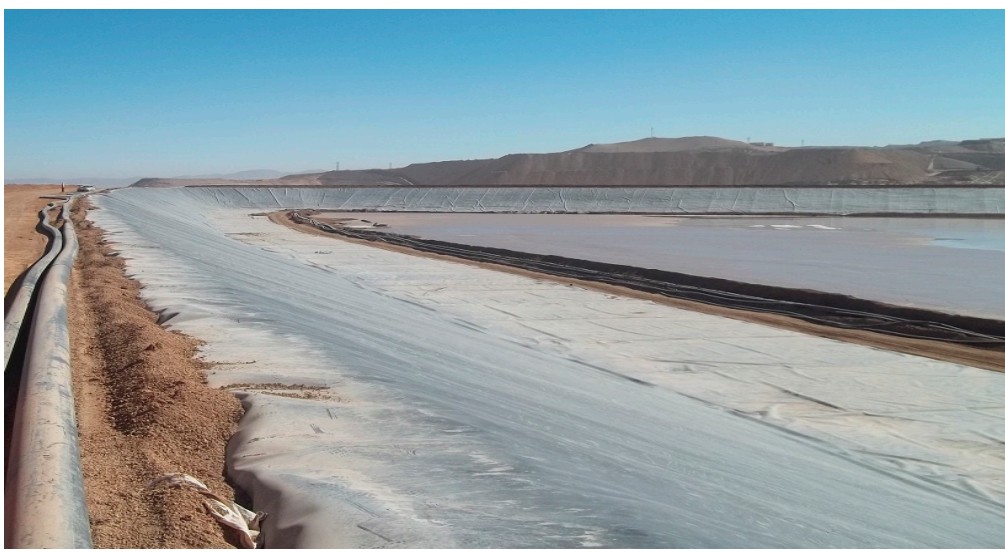

**Figure 8.** Mantos Blancos Disposal of Thickened Slimes in Tailings Storage Facility [39].

The filtered coarse tailings are transported to the filtered tailings deposit by a conveyor belt with Mobile Stacker Conveyor and Tripper. The Filtered Coarse Tailings Storage Facility is developed by the down valley placement method applied at Mobile Stacker Conveyor with Tripper forming a fan shape, where saturated filtered tailings flow along roughly repose angle slopes, buttressed at the toe by a containment dyke and downstream supported by a sedimentation collection pond. Dozers carried out spreading and compaction activities on filtered tailings lift of an average of 20–30 cm in thickness (Figures 9 and 10) [34,39].

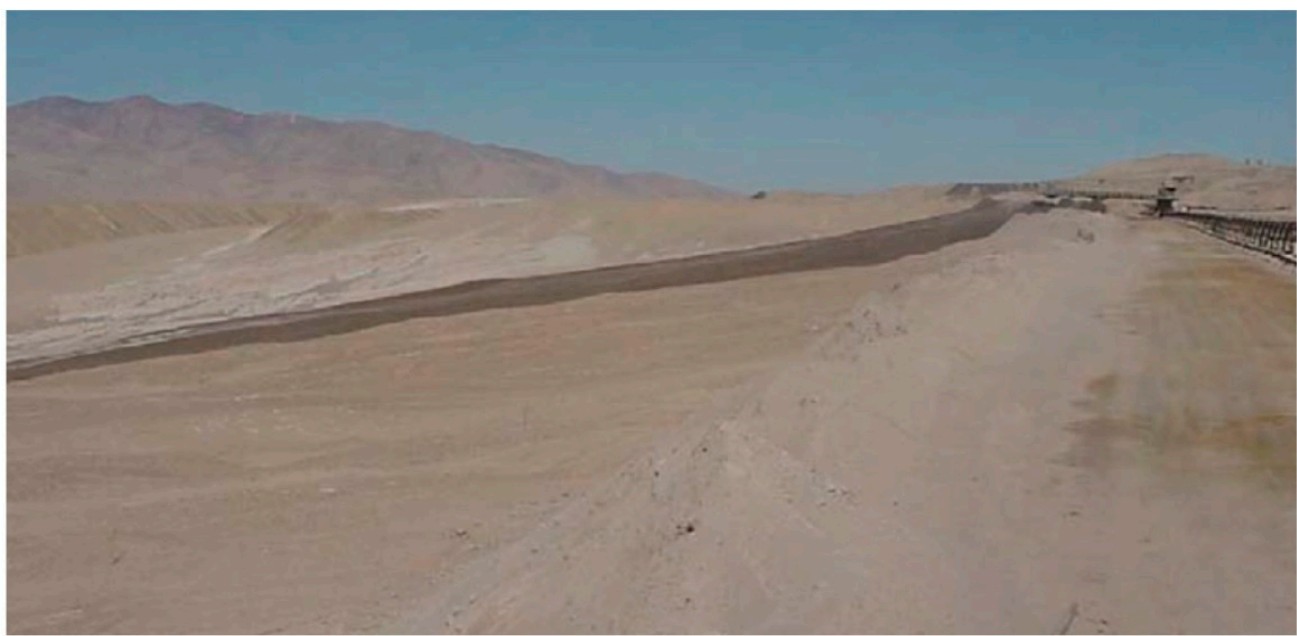

**Figure 9.** Mantos Blancos Disposal of Filtered Coarse Tailings with Mobile Stacker Conveyor and Tripper [12].

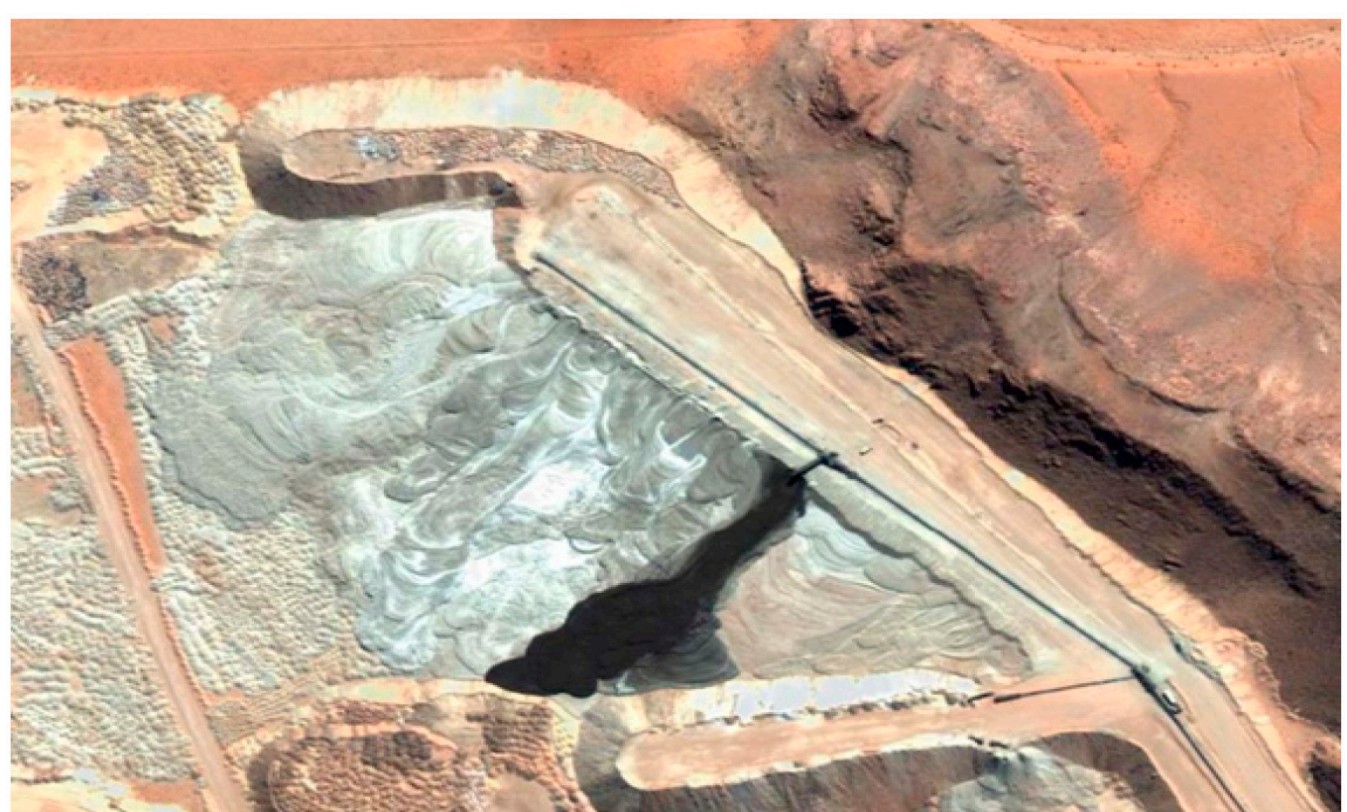

**Figure 10.** Mantos Blancos Aerial View of Disposal of Filtered Coarse Tailings [12].

## 4. Discussion: HC-HVS Dewatering Technology Application

As this case history shows, appropriate management of the dewatering of tailings can lead to better environmental management from a seepage control and water reclaim point of view. With the increase of social pressure for the safety of the TSF at the surface and environmentally friendly disposal of tailings, the disposal of filtered tailings is an option to be considered to this day.

### 4.1. Advantages (Pros)

The following paragraphs present the main advantages of tailings dewatering with the use of HC-HVS technology:

### 4.1.1. Reduction of Operating Costs

The implementation of HC-HVS units for tailings dewatering the Mantos Blancos Mine has shown to be an alternative to belt filters given their low power demand, lower operational cost, and high efficiency. This application is based on pilot tests and industrial-scale trials performed directly in the mine, which present promising results. Table 3 presents Opex Comparison between the vacuum belt filter plant and the HC-HVS plant at Mantos Blancos Project [32]:

**Table 3.** Opex Comparison between vacuum belt filter plant and HC-HVS plant at Mantos Blancos Project [32].

| Dewatering Technology Alternatives | Energy Cost kUS$/year | Operation, Maintenance, and Labour kUS$/year |
|---|---|---|
| 3 Vacuum Belt Filters of 100 m² area | 200 | 315 |
| 3 Hydrocyclone Compact Plants (HC-HVS) MUE 10/38-150.90-86 | 122 | 117 |

### 4.1.2. Performance at High Elevations

Filtering equipment suppliers have gained experience performing trials and projects during the last decade. The learning curve associated with each type of tailings differs given their particle size distribution (PSD) and the associated mineralogy. Each tailings type exhibits a unique filtering behavior for which solid/liquid separation units are required that are efficient and reliable. The performance of the dewatering of tailings by HC-HVSs is efficient at high elevations since they are not sensitive to the altitude, as is the case for vacuum filters (belt and disc filters). This characteristic is relevant depending on the project areas, such as the mountainous region of the Andes in South America.

The effect of the screens on the tailings is a consequence of the vibration and gravity. The first is a mechanical consequence of the energy applied and the second is a consequence of the mass through the force of gravity. Based on the above, the site's geographical elevation does not influence the screen's behavior since gravity is almost constant on earth, with only minor variations between sea levels and higher elevations. With this technology, atmospheric pressure, which does show important variations with elevation, does not affect the behavior of the equipment since it does not use a vacuum as part of the process.

### 4.1.3. HC-HVS Plant Location

The location of the HC-HVS plant must be determined with a trade-off study that includes factors such as (i) distance to the tailings storage facility; (ii) transport medium of the dewatered tailings; (iii) access; (iv) availability of power; and (v) requirements of maintenance and operational equipment. In general, factors (iv) and (v) indicate that this plant is located in the vicinity of the floatation plant, but this must be verified on a case-by-case basis.

### *4.2. Disadvantages (Con)*

The main disadvantages of the HC-HVS dewatering technology are presented below.

### 4.2.1. Tailings Scale Production

The filtering technologies have been applied with success for production rates of up to 20,000 tmpd, as is the case of projects at La Coipa (Chile), Chingola (Zambia), and Karara (Australia) [12]; and have shown improvements in the performance with large scale productions. In this scenario, there is still a need for more reliable equipment for large-scale filtering processes. There is a need to focus on the tailings water reclaim to improve its reuse in the mining processes. Today, the HC-HVS technology is limited in expansion because it needs more experience with production rates in the order of 10,000 mtpd to develop its full potential and show safety in the operational process of obtaining tailings "cakes", thereby improving water recovery and decreasing tailings pond water seepages.

Another relevant aspect is the definition of the screen's geometry and the filtration bed's characteristics. In practice, this is done using the initial laboratory tests but mainly with the pilot scale trials. With regards to tailings, there still is not enough experience to have a solid numeric model that allows for the dimensioning of this technology's elements as a function of the grain size and solids content of the tailings.

### 4.2.2. Fine Tailings

The nature of the tailings material is important when filtration is considered a dewatering process. Not only is the grain size of the tailings important, but also the mineralogy has a relevant role. Specifically, the high percentages of fines < 74 microns and clayey minerals tend to make the use of filtration technology difficult [15]. A relevant part of the water reclaims process in applying the HC-HVS technology, as a whole, depends on the thickening of the slimes, which implies performing an efficient liquid-solid separation with a fine fraction of the tailings. If the total tailings contain a high content of clayey minerals, the global performance of water reclaim may decrease. Attention is required to control the sedimentation and slimes thickening process.

The vibratory screens have limitations in managing fine and plastic tailings. This characteristic would eliminate the application of this technology in the majority of the tailings with a grain size in which 30% passes the #200 ASTM sieve. This restriction includes the majority of the copper tailings and all the gold tailings. However, this limitation may be overcome by the decrease of the fine fraction with the use of hydrocyclones in order to generate a product that the screens may efficiently process. Consequently, the design process first involves determining the finest grain size that the screens may dewater. Once this grain size is determined, the classification (hydrocycloning) system and the ratio or split of the coarse fraction (screen material) and a fine fraction (material to thickeners or deposit) are defined. There is an optimization stage to improve this process's performance by incorporating a fine material bed above the screen.

Another relevant property of the tailings is permeability. A medium to high permeability facilitates water removal by vibration, making the vibratory process faster. For this reason, measuring the permeability of the tailings in a laboratory is recommended. The specific weight of the particle is also relevant since, in general, a greater specific weight means a greater permeability in non-plastic tailings.

*4.3. Water Recovery Comparison with Other Mine Tailings Management Technologies*

Water losses at TSFs come from water retained in deposited tailings and the evaporation from beaches formed at the TSF. To reduce these losses, new management technologies have been developed, which seek to maximize the reclamation of water before tailings are discharged to the TSF, by cycloning, thickening, and/or filtering tailings. Improvements in conventional, thickened, paste, and filtered tailings disposal technologies need to be managed to increase water recovery and decrease water makeup (freshwater) in mining operations. These challenges have been met during the past decade in copper mining.

Table 4 shows a comparative analysis of water recovery quantities obtained with different dewatering tailings technologies assuming a tailings production of 100,000 mtpd.

**Table 4.** Water Recovery Comparison between Different Mine Tailings Management Technologies.

| Description | Unit | Conventional Tailings Management | Thickened Tailings Management | HC-HVS Tailings Management | Filtered Tailings Management |
|---|---|---|---|---|---|
| Tailings Production | mtpd | 100,000 | 100,000 | 100,000 | 100,000 |
| Cw before Dewatering | % | 28 | 28 | 28 | 28 |
| Water on Conventional Tailings | L/s | 2976 | 2976 | 2976 | 2976 |
| Cw after Dewatering | % | 50 | 60 | 70 (*) | 80 |
| Water on Dewatered Tailings | L/s | 1157 | 772 | 496 | 289 |
| Water Recovery from Dewatering Devices | L/s | 1819 | 2205 | 2480 | 2687 |
| Water Recovery from TSF | L/s | 382 | 255 | 164 | 95 |
| Total Water Recovery | L/s | 2201 | 2459 | 2644 | 2782 |
| Water Recovery Efficiency | % | 74 | 83 | 89 | 93 |

Note: The following terms mean: Cw: Tailings solid content by weight (%). (*): 70% signifies a mean target Cw value, considering dewatering tailings technologies applied.

Table 4 shows that HC-HVS technology is competitive to recover water from mine tailings, outperforming conventional tailings and thickened tailings technologies, and ranking second behind filtered tailings technology, with an efficiency of 89%. This indicates that the HC-HVS technology is an attractive option to be implemented to carry out an environmentally controlled management of mining tailings and maximize

## 5. Conclusions

Filtered tailings TSF solution today has become very competitive compared to conventional technologies because the cost of closure and post-closure are lower, the reliability of technology has increased, and the authorities and community are more prone to accept this approach considering the safety management of tailings. Under dry climate conditions,

applying the HC-HVS technology, no supernatant TSF ponds need to be monitored during operation. There is no freeboard requirement for tailings dams, which significantly reduces water losses and eliminates overtopping or piping (internal dam erosion by seepage) event risks [42,43].

The filtered dry stacked tailings case presented in this article shows that there is important potential to achieve a sustainable and cost-effective tailings management solution in desert areas such as the north of Chile and south of Peru, among others. The important benefits offered by this technology are: (i) less operational cost of the project over the mine lifetime; (ii) failure risks reduction in seismic zones improving TSF physical stability; (iii) maximum water recovery; and (iv) environmentally friendly solution are allowing smaller TSF footprints, a decrease of pond seepages, and allowing a progressive TSF reclamation.

Both the information obtained from the pilot and industrial tests from field and theory indicate that the HC-HVS technology has a promising future in dewatering tailings that are not too fine and of low plasticity. This technology presents economic and operational advantages concerning vacuum filters, particularly in sites located at high topographical elevations (masl). An important part of the mining activity in Chile is located in the Andes Mountains, at high elevations, and in an environment with scarce water availability.

To date, this technology has been applied in operations of limited production (below 6000 mtpd). Moreover, the units used have a processing capacity of 1500 mtpd. These capacities are very small to be applied in large production projects (25,000 mtpd and above) since they would require many units that would make their operation difficult, particularly in maintenance and flow conveyance. Although theory does not identify any obstacles to designing equipment of larger dimensions, it is evident that the use of larger equipment requires a thorough investigation (as was the case for press filters) or the performance of trials at an industrial scale (demonstrative plant).

In addition to the study associated with larger productions, it is particularly necessary to advance the investigation of the efficiency of this equipment and technology with fine tailings and/or with larger clay content. The above would allow for the application of this technology to iron and gold tailings, among others, marking the limits of this technology.

Over about 17 years of operation with this interesting technology, it has been established that the horizontal vibrators, together with hydrocyclones, are a viable alternative for dewatering of copper tailings in concentrator plants of midsize such 10,000 mtpd; such plants can treat tailings with a fraction of about 30–45 microns, which in many concentrators can represent the order of 60%–70% of the total mass of tailings. The true architects of the success of tailings dewater are: creating a hydrocyclone of a bed to drain the tailings; collecting the filtrate from the vibratory dewatering screens; creating a closed circuit, and the mechanical devices described here.

These benefits are a good reason for new or existing large mining operations to shift from conventional slurry tailings disposal facilities with extensive water ponds to alternative solutions with highly dewatered tailings disposal facilities. This tailings management approach is more accepted by the environmental authority and communities daily.

**Author Contributions:** Conceptualization, C.C. and E.A.; formal analysis, C.C.; investigation, C.C.; resources, E.A.; writing—original draft preparation, C.C.; writing—review and editing, C.C.; visualization, C.C.; supervision, E.A. All authors have read and agreed to the published version of the manuscript.

**Funding:** The research is funded by the Research Department of the Catholic University of Temuco, Chile.

**Data Availability Statement:** The data presented in this study are available on request from the corresponding author.

**Conflicts of Interest:** The authors declare no conflict of interest.

**Abbreviations**

| | |
|---|---|
| TSF | Tailings Storage Facility |
| BATs | Best Available Technologies |
| TTD | Thickened Tailings Disposal |
| Cw | Slurry tailings solids content by weight |
| mtpd | Metric tonnes per day |
| HC | Hydrocyclones |
| HVS | Horizontal Vibratory Screens |
| HC-HVS | Combination of Hydrocyclones and Horizontal Vibratory Screens |
| PD Pumps | Positive Displacement Pumps |
| masl | Meters above sea level |
| PU | Polyurethane |

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
