# Peer review of "An Alternative Technology to Obtain Dewatered Mine Tailings: Safe and Control Environmental Management of Filtered and Thickened Copper Mine Tailings in Chile"

_minerals, doi:10.3390/min12101334_

Round 1

Reviewer 1 Report

This is a very good paper written on a relevant topic with potential practical solutions also introduced at the same time. The language should undergo a final spelling and grammar check prior to publication. Technically, the paper is sound in that it describes actual practices of tailings management and water usage reduction in Chile.

A few editorial suggestions are provided below that should improve the paper prior to final submission.

1. The very long sentence on p. 1 that starts with "If these key..." and ends with "... for the mining industry" should be divided into two. As it currently stands, it is difficult for the reader to follow the trend of thought.

2. The use of "Cw" as an acronym is repeated throughout the text. It should be mentioned only at the first instance and then used without explaining it all over again. Currently, "Cw" is repeated multiple times with its meaning on pp. 2 (paragraphs 2, 3), 4 (last line), 5, 9, 10, 11, 12. Either 
"Cw" or "solids content" should be used but not both. The same comment can be made to the use of "HC-HVS" in paragraphs 4 and 5 on p. 4.

3. Two paragraphs on p. 4 are very close repetitions of one another. One of them should therefore be removed or modified. These are paragraphs 1 at the start of section 1.2 and part of paragraph 4 between "It is an... " and "... of copper tailings".

4. A series of 2nd- and 3rd-level headings have been automatically distorted by the formatting software and should be corrected. This seems to occur when the first word in the heading is "1st", "2nd", or a similar text indicating a numerical sequence. The affected headings are 3.2 (p. 9), 3.2.3 (p. 10), 3.3 (p. 11), and 3.3.3 (p. 11).

5. Sections 3.3.1 and 3.3.2 should be removed and replaced by a table. These are simply listing the equipment used in the process and do not convey any ideas, background, or discussions. The lists can be much better presented with a summary table.

6. The entire objective of the paper is to underscore water usage reduction that is made possible with the new technique. Hence, it would be very useful to have additional tables to 1, 2, and 3 where the percentage or volume of water reduction is indicated between the various methods used. For example, a table can be placed where the volume of water saved through paste, thickened, and HC-HVS methods are compared so that all parameters, including the cost, can be assessed.

Reviewer 2 Report

The study presented a case study about the taillings disposition, it is prepared well for the journal. The manuscript can be accepted for publication.

  • Please provide vector graphics, like .emf type images.

Reviewer 3 Report

The concept of sludge utilization by separation and dehydration of waste from mining activities in a closed cycle presented in the paper is very interesting from an economic point of view. It guarantees low costs of subsequent recovery of post-flotation wastes containing Fe, Au and other non-ferrous metal compounds. The article was prepared with great care, both in terms of graphics and content. In the opinion of the reviewer, this study meets all the requirements for publication in the journal "Minerals".
